# Small Intestine Bacterial Overgrowth Can Form an Indigenous Proinflammatory Environment in the Duodenum: A Prospective Study

**DOI:** 10.3390/microorganisms10050960

**Published:** 2022-05-02

**Authors:** Evripidis Rizos, Emmanouel Pyleris, Mark Pimentel, Konstantinos Triantafyllou, Evangelos J. Giamarellos-Bourboulis

**Affiliations:** 1Hepatogastroenterology Unit, 2nd Department of Internal Propaedeutic Medicine, Attikon University General Hospital, Medical School, National and Kapodistrian University of Athens, 124 62 Athens, Greece; evrizos@gmail.com (E.R.); ktriant@med.uoa.gr (K.T.); 2Department of Gastroenterology, Sismanogleion General Hospital, 151 26 Athens, Greece; manolis.pyleris@gmail.com; 3Medically Associated Science and Technology (MAST) Program, Cedars-Sinai Medical Center, Los Angeles, CA 90048, USA; pimentelm@cshs.org; 44th Department of Internal Medicine, Attikon University General Hospital, Medical School, National and Kapodistrian University of Athens, 1 Rimini Street, 124 62 Athens, Greece

**Keywords:** small intestinal bacterial overgrowth, inflammation, cytokines, interleukine-1b

## Abstract

Small intestinal bacterial overgrowth (SIBO) contributes to the formation of an inflammatory environment in various intestinal and extraintestinal diseases. Cytokines that participate in these mechanisms are yet to be examined. Upper gastrointestinal endoscopy with duodenal aspiration was performed in 224 patients. Quantitative cultures of aerobic species were performed, concentrations of interleukin 1β (IL-1β), interleukin 6 (IL-6), and tumor necrosis factor alpha (TNF-α) were measured, and loads of *Escherichia coli*, *Klebsiella pneumoniae*, *Methanobevibacter smithii*, and *Aeromonas* spp. were detected via real-time PCR in the duodenal fluid. Analysis showed that the odds ratio (OR) for elevated IL-1β levels was 2.61 (1.06–6.43, *p* = 0.037) among patients with SIBO compared to patients without SIBO, while there was no significant difference at elevated IL-6 and TNF-α levels between patients with and without SIBO, using ≥10³ cfu/mL as a cut-off. The presence of all three elevated cytokine levels has OR 3.47 (1.06–11.34, *p* = 0.030) among patients with SIBO. *Klebsiella pneumoniae* detection was positively related with IL-6 and TNF-α levels, when *Methanobevibacter smithii* was positively related with IL-1β levels. The presence of SIBO is associated with elevated IL-1β levels in the duodenal fluid. There is a high prevalence of all three proinflammatory cytokine levels elevated (IL-1β, IL-6, and TNF-α) in the duodenal fluid among patients with SIBO.

## 1. Introduction

Small intestine bacterial overgrowth (SIBO) represents the overgrowth of bacterial species, which usually predominate in the large bowel, in the proximal small intestine. Lactulose and hydrogen breath tests are often used as noninvasive diagnostic tests with questionable accuracy [1]. The gold-standard for diagnosis remains the culture of the fluid in the proximal part of the small intestine [2].

SIBO has been linked for many years to signs and symptoms such as bloating and diarrhea that occur due to the production of gas and other metabolites by the small intestine coliforms [3]. SIBO is more common in patients with inflammatory bowel disease (IBS) [4]. However, not all patients with SIBO present with IBS and vice versa, leading to the question if what is needed for SIBO to stimulate one IBS-like phenotype is the generation of one pro-inflammatory reaction at the level of the proximal small intestine.

Several years ago, we ran a prospective study where quantitative cultures were studied in the fluid collected from the third part of the duodenum in 897 patients who underwent upper GI tract endoscopy. In this manuscript, we present a sub-group analysis of 224 patients were the concentrations of the proinflammatory cytokines interleukin (IL)-1β, IL-6 (IL-6) and tumor necrosis factor alpha (TNF-α) were measured in the duodenal fluid.

## 2. Materials and Methods

### 2.1. Study Design

This was a prospective study conducted between September 2009 and March 2013. Study participants were patients that underwent upper gastrointestinal (GI) endoscopy in the Gastroenterology Department of Sismanogleion General Hospital of Athens, after they signed a written informed consent. Study protocol had the approval of Hospital’s Ethics Committee (Ethical Approval Number 296/2009). This study was performed in compliance with the Declaration of Helsinki and Good Clinical Practice principles.

Inclusion criteria were the following: (a) age ≥18 years; (b) written informed consent; and (c) clinical indication for upper GI endoscopy. Exclusion criteria were the following: (a) human immunodeficiency virus (HIV) infection; (b) chronic hepatitis B, hepatitis C, or hepatitis E infection; (c) Child Pugh liver cirrhosis stages B and C; (d) active upper gastrointestinal bleeding; (e) gastroesophageal reflux disease (GERD); (f) *Helicobacter Pylori* infection; (g) celiac disease; (h) enteric infections; (i) systemic sclerosis; (j) antibiotics consumption four weeks prior to endoscopy [5,6]; (k) inflammatory bowel disease; (l) microscopic colitis; (m) any gastric surgery; and (n) unstable thyroid disease.

Fluid from the third portion of the duodenum was collected during upper GI endoscopy. Water was not flushed in the duodenal lumen before the completion of aspiration. If there was inadequate volume of fluid in the duodenal lumen, there was an approach of the endoscope close to the intestinal wall, so that the biggest possible amount of fluid was aspirated. Aspirate was collected by placing a sterile suction catheter inside a sterile overtube, which was advanced through the suction channel of the endoscope. The fluid was placed immediately in transport vials and transferred to the laboratory. The samples were prospectively collected and samples were kept refrigerated until measurements were completed. Cultures of duodenal fluid and identification of bacteria were collected as described previously [7]. Briefly, an aliquot of 0.1 mL was plated onto MacConckey agar (Becton Dickinson, Conckeysville, MD, USA) and incubated for 18 h at 35 °C. Bacterial growth was determined after multiplying the number of isolated bacteria with the respective dilution factor. Concentrations of tumor necrosis factor-alpha (TNF-α), interleukin 6 (IL-6), and intereukin 1β (IL-1β) were measured in the duodenal fluid by an enzyme immunosorbent assay (R&D Inc., Minneapolis MN); the lower detection limit was 20 pg/mL for IL-1β and TNF-α and 60 pg/mL for IL-6. Quantitative real time PCR was performed in 125 patients for detecting the load of total prokaryotes, *Escherichia coli* (*E. coli*), *Klebsiella pneumoniae*, *Aeromonas* spp., and *Methanobevibacter smithii* (*M. smithii*). Quantitative PCR analyses of the above 125 patients have already been described previously [8].

For each patient enrolled the following information was recorded; age, gender, height, weight, reason for endoscopy, endoscopic findings, other diseases, and intake of any medication. All patients underwent colonoscopy in order to rule out colonic diseases such as IBD and microscopic colitis. The presence and classification of preexisting IBS was evaluated according to Rome IV criteria [9]. The presence of Helicobacter Pylori infection was evaluated with a Rapid Urease test (CLO test) during upper GI endoscopy; if positive, patient was not enrolled in the study. Furthermore, biopsies from duodenum were obtained during upper GI endoscopy in order to rule out small intestine comorbidities such as celiac disease or giardiasis.

### 2.2. Study Endpoints

The primary study endpoint was the relationship between SIBO and proinflammatory cytokines TNF-α, IL-1β, and IL-6 in the duodenal fluid. Three secondary study endpoints were set: (a) the impact of specific bacteria species that were detected via real-time PCR on the elevation of TNF-α, IL-1β, and IL-6 levels in the duodenal fluid; (b) possible risk factors that may be linked to SIBO.

### 2.3. Statistical Analysis

All statistical analyses were performed using SPSS version 20.0 for Windows software (SPSS Inc., Chicago, IL, USA). In order to explore the primary study endpoint, SIBO was defined in three different ways using three different cutoffs of concentrations of colonic type bacteria in the duodenal aspirate, i.e., >10³, >10⁴, and >10⁵ cfu/mL. Comparisons for qualitative variables between patients with SIBO and those without SIBO were calculated by the Chi-square test. Cytokines were expressed as % of patients with detectable concentrations. Variables with normal distribution of values were compared using Student’s *t*-test and variables with non-normal distribution of values were compared using Mann–Whitney U test. Odds ratios (OR) and 95% confidence intervals (CI) were calculated by Mantel–Haenszel’s statistics. Step-wise forward logistic regression analysis was performed with the presence of SIBO as a dependent variable and the presence of IBS or other co-morbidities, drug intake, findings from endoscopy, and cytokine levels as independent variables. *p* value less than 0.05 was considered significant.

## 3. Results

Demographic characteristics of enrolled patients are classified into those with and those without SIBO and are shown in Table 1. As a whole, patients with SIBO were older, had a higher frequency of type 2 diabetes mellitus and predominant-diarrhea IBS.

### 3.1. Primary Endpoint

An analysis between the presence of SIBO and the presence of elevated proinflammatory cytokine levels in the duodenal fluid was conducted, where all diagnostic cut-offs of isolation of bacteria from the duodenal aspirate were considered as previously discussed [2], i.e., ≥10³ cfu/mL, ≥10⁴ cfu/mL, and ≥10⁵ cfu/mL. Regardless of the set cut-off, the OR for elevated IL-1β levels was greater among patients with SIBO compared to patients without SIBO. On the contrary, there was no significant difference at elevated IL-6 and TNF-α levels between patients with and without SIBO (Table 2 and Table 3). The presence of all three elevated cytokine levels has OR 3.47 (1.06–11.34, *p* = 0.03) among patients with SIBO (Figure 1). No significant association between the number of elevated cytokine levels and the presence of IBS was detected (Figure 1).

### 3.2. Secondary Endpoints

PCR counts of *Escherichia coli*, *Klebsiella pneumoniae*, *Aeromonas* spp., and *Methanobevibacter smithii* were measured in the duodenal fluid of 125 patients, and analysis investigated the relationship with cytokine levels. Analysis with the Mann–Whitney U test showed that *Klebsiella pneumoniae* PCR copies were positively related to IL-6 and TNF-α levels, while *M. smithii* PCR copies were positively related with IL-1β levels (Table 4). PCR copies of *E. coli* and *Aeromonas* spp. were not related to cytokine levels in the duodenal fluid (Figure 2, Figure 3 and Figure 4).

## 4. Discussion

Our study investigates the connection between SIBO and proinflammatory cytokines’ levels in the duodenal fluid. The inflammatory substrate of SIBO and the contribution of cytokines in its interactions have been studied for decades, and there is an assumption that the presence of SIBO contributes towards the upregulation of inflammatory cytokines, mainly in the intestine, and systematically at a lesser point. Cytokine mRNA expression of German shepherd dogs with SIBO was investigated but no significant difference was detected [10]. IL-1β, IL-6, Toll-like receptor 4 (TLR-4), and interleukin 10 (IL-10) levels were measured in mucosal tissue of ileum in mice with post infectious IBS, and TLR-4 was detected at higher levels in mice with SIBO [11]. TLR-4 expression from liver biopsies in patients with metabolic associated fat liver disease (MAFLD) was also higher when SIBO was present, although no significant difference at serum TNF-α levels was detected [12]. Serum IL-6, IL-8, and TNF-α levels were significantly higher in SIBO positive ulcerative colitis patients as compared to SIBO negative patients [13]. SIBO rates were higher in patients with Crohn’s disease and led to elevated levels of fecal calprotectin, a well-known biomarker indicating local inflammation in small intestine [14]. Riordan et al. studied mucosal cytokine production in SIBO and also demonstrated no difference in TNF-α along with interferone-γ levels, while detected levels of IL-6 were higher in SIBO positive patients [15]. No differences in serum TNF-α, IL-6, and IL-8 levels were observed between SIBO positive and SIBO negative IBS patients, whereas levels of anti-inflammatory cytokine IL-10 were lower in the first group [16]. Similar proinflammatory patterns with elevated cytokines that also led to increased small bowel homing T-cells were detected in patients with predominant diarrhea IBS [17] and with *Helicobacter pylori* negative functional dyspepsia [18], both of the above mentioned conditions closely linked to SIBO. Our study detected a positive connection between SIBO per se and the presence of an indigenous proinflammatory environment in the small bowel.

Although IL-1β has an established role in pathophysiology of inflammatory bowel disease (IBD) and irritable bowel syndrome (IBS) [19], studies on changes in IL-1β levels in patients with SIBO are rare. Patients with IBS were proven to have higher small intestine mucosal IL-1β levels when they were SIBO positive [20]. In the same study, increased IL-1β level was predominantly associated with bloating and loose stools (Bristol type 6). In a recent study, also on patients with IBS, IL-1β levels in the peripheral blood were higher in the SIBO positive group compared to the SIBO negative group [21]. IL-1β levels decreased significantly after antibiotic treatment strengthening the hypothesis of SIBO’s contribution to elevated IL-1β. For the first time in our study the connection between SIBO and elevated intestinal IL-1β levels was examined, regardless of the presence of other small intestine disorders.

Recent studies suggest that distinct commensals such as *Proteus mirabilis* and *Escherichia coli* can cause IL-1β dependent intestinal inflammation via NLRP3 inflammasome activation in IBD models [22,23]. NLRP3 inflammasome is a well-defined intracellural multiprotein complex found in monocytes that plays an important role in intestinal homeostasis by regulating not only IL-1β activation, but also IL-18 activation and pyroptosis, among others [24]. In this context, SIBO presence could lead to NLRP3 inflammasome dysfunction and intestinal inflammation due to consequent IL-1β upregulation. Further research is needed to explore a possible dysfunction of NLRP3 inflammasome in patients with SIBO.

The proinflammatory pattern with all three cytokines (IL-1β, IL-6, and TNF-α) elevated in the duodenal fluid that was strongly connected with SIBO in our study may suggest an intestinal motility disruption mechanism. IL-1β is considered as a mediator of intestinal dysfunction as it causes a decrease in smooth muscle contractility in rat ileum [25]. Selective jejunal manipulation causes a decrease in smooth muscle contractility and significant delay in colonic transit is accompanied by upregulation of IL-6 and TNF-α [26]. TNF-α, in turn, directly induces motor dysfunctions by acting on the smooth muscle, as shown in a trinitrobenzenesulfonic acid (TNBS) induced colitis model in mice [27]. Small bowel motility is the most important mechanism preventing SIBO. Disruption of the enteric nervous system, visceral musculature, or both, can result in SIBO, and SIBO in turn can aggravate these impaired functions [28]. Indigenous proinflammatory cytokines can play an important factor in this bidirectional connection.

*Klebsiella pneumoniae* is a widely known pathogen that inhabits the large bowel and is shown to have high frequency in SIBO [7,29]. Our study shows that high PCR counts of *Klebsiella pneumoniae* in the duodenal fluid are connected with elevated IL-6 and TNF-α levels. *Methanobevibacter smithii*, an archaea that is considered to be the most abundant methanogen in the human gut, has been positively linked to predominant constipation IBS as excessive methane production in the small intestine leads to slower small bowel transit and increased contractile activity [30]. Our study shows that high PCR load of *M. smithii* is associated with elevated IL-1β levels in the duodenal fluid. Considering that IL-1β may be a contributing factor in intestinal motility dysfunction [18], *M. smithii* could possibly lead to local small intestine immune activation through IL-1β upregulation.

The presence of SIBO was significantly more frequent among patients with type 2 diabetes mellitus (T2DM) in our study. This finding comes in accordance to previous literature that shows a high prevalence of SIBO in patients with T2DM, together with delayed intestinal transit [31]. As T2DM can lead to visceral neuropathy and slowed orocecal transit [32], an assumption could be made that T2DM can lead to SIBO through impaired intestinal motility. However, a recent study indicates that SIBO positive patients with T2DM have elevated serum proinflammatory cytokine (TNF-α, IL-6, and IL-10) levels implying a bidirectional connection between T2DM and SIBO [33]. As previously discussed, SIBO may also promote local inflammation in patients with MALFD [12], another aspect of metabolic syndrome, strengthening the hypothesis of a linkage between SIBO and metabolic syndrome that can lead to both intestinal and systemic inflammation.

Furthermore, our study showed that patients with a BMI lower than 22 have a higher risk of SIBO, as shown in Table 2 and Table 3. Taking into consideration SIBO’s classic symptoms are malnutrition and weight loss, among others, it should come as no surprise that there is a higher prevalence of SIBO among underweight patients. Equivalent findings presented among patients with chronic pancreatitis and pancreatic exocrine insufficiency, where patients with weight loss were more likely to have SIBO [34]. On the other hand, no significant relationship between weight loss and SIBO was found in patients with Parkinson’s disease [35]. Further studies are needed to support this association. In addition, PPI intake did not affect SIBO prevalence among patients, which comes into accordance with a previous study of a population of 897 patients where neither the intake nor the duration of PPI consumption was found to have any impact on SIBO [4].

Our study presents certain limitations; first is the non-randomized design. Second is the lack of use of a method to prevent contamination during endoscopy, although there was a low probability of contamination, as the detected aerobe bacteria were usually inhabitants of the large bowel. Third is the absence of concomitant hydrogen breath test that would rule out the probability of possible contamination. Fourth is the lack of anaerobic cultures. Last is the relatively small number of SIBO positive patients.

## 5. Conclusions

Our study prospectively demonstrated, for the first time, the association of elevated IL-6, IL-1β, and TNF-α levels combined and IL-1β levels alone at a lesser point in the duodenal fluid with SIBO. The study also demonstrated that the presence of *M. smithii* is related with elevated IL-1β levels, while the presence of *Klebsiella pneumoniae* is related with high IL-6 and TNF-α levels in the duodenal fluid.

## Figures and Tables

**Figure 1 microorganisms-10-00960-f001:**
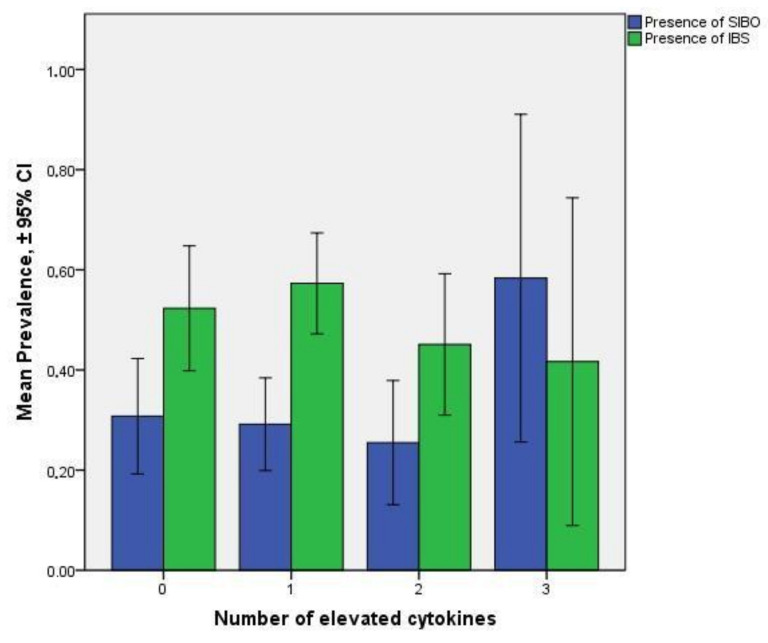
Prevalence of SIBO (using ≥10³ cfu/mL as a cut-off) (blue bars) and IBS (green bars) among patients with 0, 1, 2, and 3 elevated cytokine levels (TNF-α, IL-6, and IL-1β) detected in the duodenal fluid. Patients with all 3 cytokine levels elevated have OR 3.47 (1.06–11.34, *p* = 0.03) presenting with SIBO.

**Figure 2 microorganisms-10-00960-f002:**
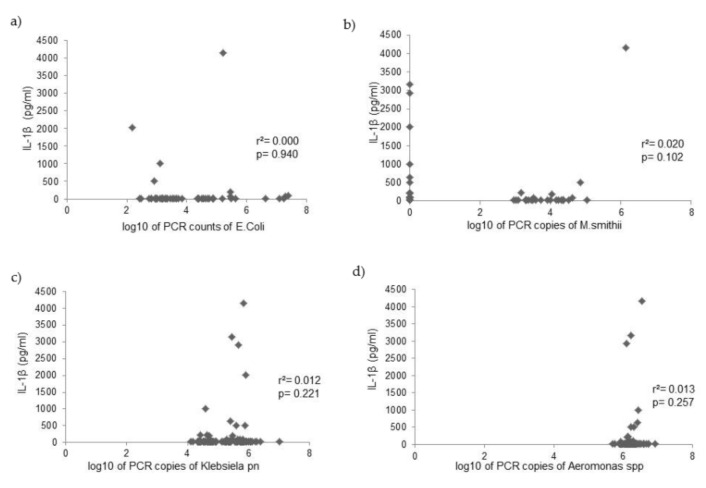
Correlation between IL-1β levels and PCR copies of (**a**) *Escherichia coli*, (**b**) *Methanobevribacter smithii*, (**c**) *Klebsiella pneumoniae*, and (**d**) *Aeromonas* spp. in the duodenal fluid.

**Figure 3 microorganisms-10-00960-f003:**
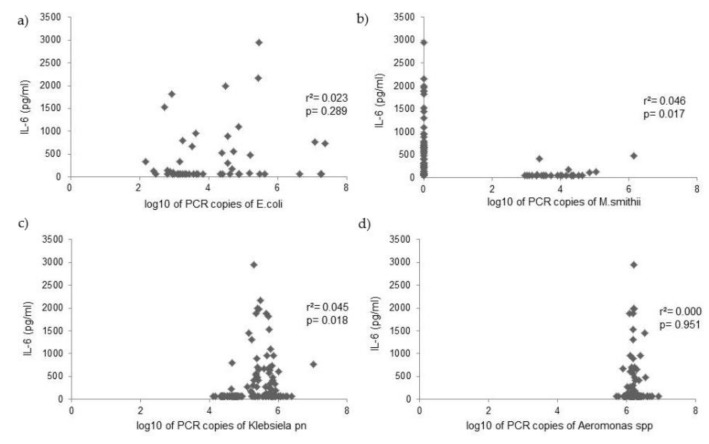
Correlation between IL-6 levels and PCR copies of (**a**) *Escherichia coli*, (**b**) *Methanobevribacter smithii*, (**c**) *Klebsiella pneumoniae*, and (**d**) *Aeromonas* spp. in the duodenal fluid.

**Figure 4 microorganisms-10-00960-f004:**
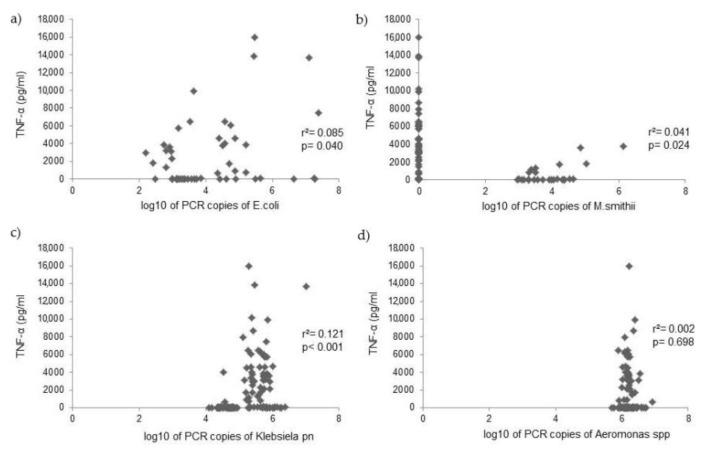
Correlation between TNF-α levels and PCR copies of (**a**) *Escherichia coli*, (**b**) *Methanobevribacter smithii*, (**c**) *Klebsiella pneumoniae*, and (**d**) *Aeromonas* spp. in the duodenal fluid.

**Table 1 microorganisms-10-00960-t001:** Demographic characteristic of enrolled patients.

	No SIBO (*n* = 156)	SIBO (*n* = 68)	*p*
Male (*n*, %)	73 (46.8)	36 (52.9)	0.397
Age ≥ 60 years (*n*, %)	93 (59.6)	55 (80.9)	0.002
BMI ≥ 22 kg/m^2^ (*n*, %)	136 (87.2)	52 (76.5)	0.045
Presence of IBS (*n*, %)	75 (48.1)	42 (61.8)	0.059
Type of IBS (*n*, %)			
IBS-D	23 (14.7)	18 (26.5)	0.037
IBS-C	13 (8.3)	3 (4.4)	0.295
Mixed type IBS	39 (25.0)	21 (30.9)	0.361
Comorbidities (*n*, %)			
T2DM	28 (17.9)	21 (30.9)	0.036
CHF	36 (23.1)	20 (29.4)	0.314
COPD	16 (10.3)	5 (7.5)	0.513
CRF	7 (4.5)	2 (2.9)	0.588
Solid tumor malignancy	14 (9.0)	4 (5.9)	0.434
History of drug intake (*n*, %)			
PPIs	38 (24.4)	13 (19.1)	0.390
NSAIDs	9 (5.8)	2 (2.9)	0.368
Aspirin	27 (17.3)	15 (22.1)	0.402
Acenocumarone	11 (7.1)	7 (10.3)	0.412
H2-blockers	2 (1.3)	1 (1.5)	0.910
Antacids	5 (3.2)	1 (1.5)	0.460
Clinical reason for gastroscopy (*n*, %)			
Dyspepsia	94 (60.3)	39 (57.4)	0.684
Anemia	73 (46.8)	36 (52.9)	0.397
Fever of unknown origin	7 (4.5)	2 (2.9)	0.588
Endoscopic findings (*n*, %)			
Gastritis	69 (44.2)	25 (36.8)	0.298
Duodenal ulcer	12 (7.7)	8 (11.8)	0.326
Gastric ulcer	2 (1.3)	1 (1.5)	0.910

**Table 2 microorganisms-10-00960-t002:** Univariate analysis of independent factors and their relationship with SIB.

	OR	95% Cis	*p*
Male gender	0.504	0.167–1.522	0.224
Age ≥ 60 years	16.264	2.917–90.684	0.001
BMI ≥ 22	0.269	0.059–1.232	0.091
Obesity	1.466	0.349–6.151	0.601
IBS	8.964	2.716–29.581	<0.001
T2DM	2.927	0.832–10.295	0.094
CHF	0.961	0.238–3.869	0.955
COPD	0.312	0.060–1.632	0.168
CRF	0.596	0.043–8.243	0.700
Solid Tumor Malignancy	0.283	0.038–2.105	0.218
PPIs	0.382	0.098–1.498	0.168
NSAIDs	0.682	0.031–14.928	0.808
Aspirin	1.952	0.403–9.452	0.406
Acenocumarone	1.236	0.168–9.069	0.835
H2 blockers	n/a	0.000	1.000
Antacids	1.813	0.114–28.959	0.674
Dyspepsia	0.527	0.160–1.732	0.291
Anemia	0.641	0.160–2.568	0.530
Fever of unkown origin	1.354	0.109–16.771	0.813
Gastritis	1.527	0.515–4.526	0.445
Duodenal Ulcer	1.471	0.242–8.931	0.675
Gastric Ulcer	<0.001	0.000	0.999
Elevated IL-1β	3.803	0.953–15.171	0.058
Elevated IL-6	2.099	0.608–7.252	0.241
Elevated TNFa	0.381	0.094–1.542	0.176

**Table 3 microorganisms-10-00960-t003:** Multivariate forward stepwise logistic regression analysis of statistically significant factors independently related with SIBO.

	OR	95% Cis	*p*
Age ≥ 60 years	4.10	1.93–8.70	<0.001
BMI ≥ 22	0.37	0.17–0.83	0.016
IBS	2.10	1.12–3.96	0.021
Elevated IL-1β	2.61	1.06–6.43	0.037

**Table 4 microorganisms-10-00960-t004:** Linkage between elevated cytokine levels and PCR copies of bacterial species, and bacterial counts in aerobic cultures, using Mann–Whitney U test.

	IL-1β (−)	IL-1β (+)	*p*	IL-6 (−)	IL-6 (+)	*p*	TNF-α (−)	TNF-α (+)	*p*
	Mean Rank	Mean Rank		Mean Rank	Mean Rank		Mean Rank	Mean Rank	
*E. coli*	62.4	66.5	0.603	59.2	68.3	0.118	62.4	63.3	0.883
*Klebsiella pn.*	62.1	68.0	0.511	54.1	75.6	0.001	50.1	68.4	0.010
*M. smithii*	61.0	74.0	0.044	66.7	57.7	0.053	67.4	61.2	0.217
*Aeromonas* spp.	63.3	61.1	0.801	65.6	59.4	0.110	67.6	61.1	0.353
*Bacterial counts* *in aerobic cultures*	110.4	129.0	0.099	112.1	113.8	0.829	111.0	113.2	0.768

## Data Availability

Data will be submitted upon request.

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
