# Peer review of "Small Intestine Bacterial Overgrowth Can Form an Indigenous Proinflammatory Environment in the Duodenum: A Prospective Study"

_microorganisms, 2022, doi:10.3390/microorganisms10050960_

Round 1

Reviewer 1 Report

The article focuses on SIBO diagnosis by duodenal juice sampling and correlates PCR-based bacterial populations with several cytokines and demographics. Manuscript is generally well written and the rational is interesting. There are however, several concerns which are listed below. I would be happy to see the addressed point in the revised paper.

  • Please clarify how this study can be prospective and being conducted ten years ago
  • Ethical approval number is required
  • Please mention SIBO profile of patients since several factors favour the disease beyond systemic sclerosis, eg operations, long-term PPI intake with associated dysbiosis etc
  • You mention IBD as exclusion criterion, which is correct. Did you perform colonoscopies, or just from the history taking? It would be reasonable to state, that you also excluded microscopic colitis patients (particularly since many of them where under NSAIDs or PPI, which predispose to it. SSRI?)
  • Helicobacter pylori Status as a central key-player of homeostasis and dysbiosis was not checked, it should be mentioned to the limitations
  • Child Pugh stage: please change 2,3 to B and C, respectively
  • If immune compromized patients were included, HEV should be also ruled out
  • Since I assume you also obtained routine duodenal biopsies, it would be reasonable to document and state, that the eligible patients did not have any of other morbidities, like sprue, giardiasis, autoimmune enteropathy etc
  • Primary endpoint: “where all diagnostic cut-offs of 113 isolation of bacteria from the duodenal aspirate were considered ie ≥10³ cfu/ml, …” please cite if this cut-off is validated or mention if its arbitrary
  • Regarding IBS, please explain how IBS was diagnosed, since it is a diagnosis of exclusion. It would be reasonable to state, that IBS was preexisting, diagnosed e.g. by means of Rome III/IV criteria and later on it was complicated with SIBO
  • Names of bacteria are not universally in italics, please apply accordingly
  • A brief statement regarding Helsinki principles and GCP should be added
  • I would have expected, since IL-1beta is measured, a connection and further investigation of inflammasome mechanism with pyroptosis
  • Since contamination is a possible issue, which the study cannot examine, it would be ideally combined with an H2 breath test, which was not done. This should be added as a further limitation
  • Please change NAFLD to MAFLD. Abbreviation is actually not needed, since it is not mentioned a second time throughout the manuscript
  • Make an effort to update some of your references which are outdated

Author Response

To the Editor

Microorganisms

Dear Sir or Madam,

We submit for review and possible publication the revision of our manuscript entitled “Small intestine bacterial overgrowth can form an indigenous proinflammatory environment in the duodenum: a prospective study” according to the comments of the two reviewers. Below you may find our point-by-point responses to the comments. We hope that the revised manuscript will meet the requirements for publication in your prestigious Journal.

REVIEWER 1:

  • Please clarify how this study can be prospective and being conducted ten years ago

Response: The samples were prospectively collected and samples were kept refrigerated until measurements were done. This is fully clarified on page 2, paragraph 4 of the revised manuscript.

  • Ethical approval number is required

Response: The code of the approval is 296/2009. This is now provided on page 2, paragraph 2 of the revised manuscript.

  • Please mention SIBO profile of patients since several factors favour the disease beyond systemic sclerosis, eg operations, long-term PPI intake with associated dysbiosis etc

Response: We want to thank the reviewer for the valuable comment. Possible confounding factors such as liver cirrhosis or inflammatory bowel disease had been set by the investigators as exclusion criteria. An analysis of the impact of PPI intake on the prevalence of SIBO in the entire study population of 897 patients has already been published and referenced as reference 4 in the list of references. Neither the intake of PPIs nor the duration of intake was found to have any impact on SIBO. This is now mentioned on page 10, paragraph 1 of the revised manuscript.

  • You mention IBD as exclusion criterion, which is correct. Did you perform colonoscopies, or just from the history taking? It would be reasonable to state, that you also excluded microscopic colitis patients (particularly since many of them where under NSAIDs or PPI, which predispose to it. SSRI?)

Response: All these patients underwent colonoscopy. This is now mentioned on page 2, paragraph 5 of the revised manuscript.

  • Helicobacter pylori status as a central key-player of homeostasis and dysbiosis was not checked, it should be mentioned to the limitations

Response: CLO test was performed in all patients. If positive, they were not enrolled in the study. This is now mentioned on page, paragraph 5 of the revised manuscript.

  • Child Pugh stage: please change 2,3 to B and C, respectively

Response: Thank you for the valuable remark. We corrected as requested in the “Study Design” section (page 2, line 62).

  • If immune compromized patients were included, HEV should be also ruled out

Response: Thank you for your comment. This has now been added in our exclusion criteria (page 2, line 59)

  • Since I assume you also obtained routine duodenal biopsies, it would be reasonable to document and state, that the eligible patients did not have any of other morbidities, like sprue, giardiasis, autoimmune enteropathy etc

Response: Biopsies were negative for any of these diseases. This has now been added on page, paragraph 5 of the revised manuscript.

  • Primary endpoint: “where all diagnostic cut-offs of 113 isolation of bacteria from the duodenal aspirate were considered ie ≥10³ cfu/ml, …” please cite if this cut-off is validated or mention if its arbitrary

Response: Thank you for your valuable comment. These are validated cut-offs based on the latest North American Consensus. This reference is now cited as seen on Results, “Primary Endpoint” section (page 3, line 115).

  • Regarding IBS, please explain how IBS was diagnosed, since it is a diagnosis of exclusion. It would be reasonable to state, that IBS was preexisting, diagnosed e.g. by means of Rome III/IV criteria and later on it was complicated with SIBO

Response: We want to thank the reviewer for the remark. We state in “Study Design” section that “…Presence and classification of IBS was recorded according to Rome criteria”. We amend that diagnosis of IBS was preexisting before patients underwent upper GI endoscopy, was made according Rome IV criteria and cite the appropriate reference (page 2, line 83).

  • Names of bacteria are not universally in italics, please apply accordingly

Response: Thank you for the remark. We amend all the names of bacteria throughout the text in italics.

  • A brief statement regarding Helsinki principles and GCP should be added

Response: Thank you for the comment. Our study protocol was approved by Sismanogleion General Hospital’s Ethical Committee as already stated and of course Good Clinical Practice principles are of utmost importance to us. We added a brief statement in “Study Design” section (page 2, line 56).

  • I would have expected, since IL-1beta is measured, a connection and further investigation of inflammasome mechanism with pyroptosis

Response: We want to thank the reviewer for the insightful comment. Inflammasomes and especially NLRP3 inflammasome play a main regulative role not only in IL-1β activation, but also in IL-18 activation and pyroptosis, among others. Our team has been involved in NLRP3 inflammasome role especially in IBD patients (Lazaridis et al., Activation of NLRP3 Inflammasome in Inflammatory Bowel Disease: Differences Between Crohn's Disease and Ulcerative Colitis, Dig Dis Sci. 2017). Unfortunately, our study design did not encompass any ways of direct or indirect detection for inflammasomes activation, apart from IL-1β levels measurement. We added a small paragraph in “Discussion” section where we propose possible pathways including NLRP3 inflammasome and highlighting the need for further studies (page 8, line 202).

  • Since contamination is a possible issue, which the study cannot examine, it would be ideally combined with an H2 breath test, which was not done. This should be added as a further limitation

Response: Thank you for your valuable comment. We amended the paragraph about limitations in “Discussion” section according to your suggestion (page 9, line 242).

  • Please change NAFLD to MAFLD. Abbreviation is actually not needed, since it is not mentioned a second time throughout the manuscript

Response: We want to thank the reviewer for the insightful annotation. Although it is still proposed, MAFLD is a more updated and appropriate term, so we corrected the script twice in “Discussion” section.

  • Make an effort to update some of your references which are outdated

Response: Thank you for the comment. Over half of our references are published within the last 5 years. Older references are landmark studies with many citations to date. We replaced reference number 9 with an updated one.

Reviewer 2 Report

In the present experimental study Rizos et al found that SIBO is associated with higher pro-inflammatory cytokine levels (such as IL 1b) in duodenal aspirate fluid. Main comments:

  1. A linguistic revision is necessary. For example, lines 66-67 are hard to understand
  2. An increase of pro-inflammatory cytokines in the duodenum may be secondary also to other conditions such as H. Pylori, other enteric infections or parasites (Giardia), which seem not to have been ruled out (lines 57-63). Please explain.
  3. OR is an incorrect way to express a comparison between continuous variables between groups (lines 21-22). In this case t test or Mann-Whitney U are more proper.
  4. Lines 236-242: this sentence is not in agreement with other evidences showing that obese patients carry a higher risk of SIBO (see Ierardi E et al, Scand J Gastroenterol 2016). Please discuss.
  5. Figure 1 caption is hard to understand. Moreover, the OR = 3.47 is in reference to what? IBS or controls?

Author Response

To the Editor

Microorganisms

Dear Sir or Madam,

We submit for review and possible publication the revision of our manuscript entitled “Small intestine bacterial overgrowth can form an indigenous proinflammatory environment in the duodenum: a prospective study” according to the comments of the two reviewers. Below you may find our point-by-point responses to the comments. We hope that the revised manuscript will meet the requirements for publication in your prestigious Journal.

REVIEWER 2:

  • A linguistic revision is necessary. For example, lines 66-67 are hard to understand

Response:  Thank you for the comment. The above mentioned sentence was re-written.

  • An increase of pro-inflammatory cytokines in the duodenum may be secondary also to other conditions such as H. Pylori, other enteric infections or parasites (Giardia), which seem not to have been ruled out (lines 57-63). Please explain.

Response: CLO test was performed in all patients. If positive, they were not enrolled in the study. Also, biopsies from duodenum were obtained during upper GI endoscopy in order to rule out small intestine morbidities. This is now mentioned on page 2, paragraph 5 of the revised manuscript.

  • OR is an incorrect way to express a comparison between continuous variables between groups (lines 21-22). In this case t test or Mann-Whitney U are more proper.

Response: Thank you for the comment. As seen in both “Study Design” (page 2, line 77) and “Figure 1” any detectable levels of cytokines above the lower detection limit are considered elevated levels. In this case, cytokine levels are split into 2 categories as a qualitative variable and OR is an appropriate comparison method. We amended the above mentioned sections so that this can seem more interpretable.

  • Lines 236-242: this sentence is not in agreement with other evidences showing that obese patients carry a higher risk of SIBO (see Ierardi E et al, Scand J Gastroenterol 2016). Please discuss.

Response: We now cite (page 8, line 232) that metabolic syndrome has positive association with SIBO.

  • Figure 1 caption is hard to understand. Moreover, the OR = 3.47 is in reference to what? IBS or controls?

Response: Thank you for the comment. As mentioned in “Primary Endpoint” section presence of all three elevated cytokine levels has OR 3.47 (1.06 – 11.34, p= 0.03) among patients with SIBO compared to patients without SIBO. We take the opportunity after your valuable remark to modify Figure 1 caption in a more articulate form.